# Parents' Views on Family Resiliency in Sustainable Remote Schooling during the COVID-19 Outbreak in Finland

**Teija Koskela** [1,*] , **Kaisa Pihlainen** [2] , **Satu Piispa-Hakala** [2] , **Riitta Vornanen** [3]
**and Juha Hämäläinen** [3,4,5]

1. Faculty of Education, University of Turku, 26100 Rauma, Finland
2. Philosophical Faculty, University of Eastern Finland, 80100 Joensuu, Finland; kaisa.pihlainen@uef.fi (K.P.); satu.piispa-hakala@uef.fi (S.P.-H.)
3. Faculty of Social Sciences and Business Studies, University of Eastern Finland, 70210 Kuopio, Finland; riitta.h.vornanen@uef.fi (R.V.); juha.hamalainen@uef.fi (J.H.)
4. Faculty of Social Studies, University of Ostrava, 70200 Ostrava, Czech Republic
5. School of Social Development and Public Policy, Fudan University, Shanghai 200433, China
* Correspondence: teija.koskela@utu.fi

**Abstract:** The closure of schools because of the COVID-19 pandemic created a challenge for families and teachers in supporting children's remote schooling. This study investigates parents' perspectives on their accommodation to the rapid change to remote schooling from the point of view of sustainable education. The study was conducted at the beginning of the COVID-19 pandemic in spring 2020 via an online questionnaire for parents, to which 316 voluntary participants responded. Data were analyzed using a theory-driven content analysis. According to the results, parents were worried about the learning and wellbeing of their children as well as management of daily life and use of information and communications technology (ICT). The results show the importance of schools and teachers as well as networks in supporting family resilience during rapid changes. Families' individual needs should be acknowledged and met in a sustainable way to support children's learning in changing settings, including remote schooling.

**Keywords:** remote schooling; sustainable society; inclusive education; parents; resilience; COVID-19 pandemic

## 1. Introduction

The education system has been subjected to a stress test because of the COVID-19 pandemic. This research contributes to the discussion by asking how families coped during the pandemic and remote learning with their children in Finland. The Global Education Monitoring Report (GEM) highlights the crucial role of education in the process of enhancing the sustainable development goals (SDGs) [1]. In this study, we understand the concept of sustainability according to the framework of the 2030 Agenda for Sustainable Development [2], especially emphasizing goal number four, quality education. This goal follows guidelines written in the Incheon Declaration [2] to ensure inclusive and equitable quality education and promote lifelong learning opportunities for all. No one should be left behind.

The GEM [1] underlines the role of education in human wellbeing. It connects education to intergenerational equity and justice. According to the report, education is responsible for raising children to be healthy and resilient. The concept of resilience is mentioned on the one hand as a capability of an individual, and on the other hand as a quality of the social and material environment.

"Education can give people the skills to participate in shaping and maintaining more sustainable cities, and to achieve resilience in disaster situations." [1]

During exceptional times, such as the COVID-19 pandemic, education in the Nordic welfare countries calls for examination. The crisis has shown us the limits of the ordinarily, well-run educational system. The Finnish educational system is internationally recognized for its inclusiveness and equity. The pandemic and remote schooling constituted a stress test for the system, and they revealed some fragilities. In this article, we use the COVID-19 pandemic as a lens to observe the Finnish educational system at a time of emergency. Even though the Finnish school system aims at equality, there were some issues with remote schooling as well as information and communications technology (ICT) that increased inequality and might have lowered the quality of education technology provided by homes and schools, internet access, and ICT competence of teachers. We focused especially on the family's point of view. The crucial question is how the educational system seemed to work from the family's perspective during the COVID-19 pandemic in 2020 immediately after Finnish schools were closed. On 13 March, 2020, the Finnish government instructed all schools to close on March 18 to restrict the spread of the pandemic.

## 1.1. The Finnish System of Welfare and Education

As Finnish society is organized according to the Nordic welfare model, school policy, child welfare policies, and the system of child and family services of the country embody this societal order. Welfare is essentially based on wide public responsibility for promoting social security through a comprehensive system of benefits and services for all citizens. The child welfare policy embodies this ideology with wide legal rights for children and an extensive system of welfare services [3,4]. Accordingly, there is a comprehensive statutory mechanism of school welfare work aiming to provide psychological, social, and health care services for students according to their needs in cooperation with parents [5]. Thus, the shift from normal attendance at school to remote learning not only affected students' learning, but also distanced them from school welfare services.

The Nordic welfare concept is essentially normative, instantiating high ethical values such as equality, social justice, and social security. Schools intend to implement corresponding values, norms, and principles. The ideas of social inclusion and alleviation of social exclusion are fundamental elements of school policy and practice as well as child welfare in Finland. By definition, exclusion and inclusion are complicated multidimensional phenomena to measure [6]. In a school context, inclusion, as opposed to segregation, indicates policies and practices to support children's studying in common groups by versatile means. As a paradigm of school pedagogy, inclusion refers to teachers' sensitivity to recognizing and embracing diversity [7] and commitment to responding to the needs of diverse learners [8]. The purpose is to support the idea of full participation of all pupils in education [9] in "school for all" [10]. Inclusive educational collaboration with families is a critical element, and it aims to share decision-making [11] and enhance mutual understanding about families' living contexts and values [12]. Trust, collaboration, and wellbeing have been identified as key elements of the Finnish school culture [13].

The school, in fact, is deeply bound up with all other education and welfare institutions of society in fulfilling the mission of child welfare according to the wide concept of child welfare policy. This concept defines the school's attentiveness to the mission of child welfare as an integrated part of the system of legal rights of the child, obliging all professionals who work with children and families. So far, there is some tentative research-based information about the influences of remote schooling on practices of teaching that is not yet published in international journals under the control of the scientific community, but there is very little research on the effect of the special arrangements of school practices caused by the COVID-19 pandemic on children's and families' everyday lives and wellbeing. Concerning the influence on teaching and learning, many inequalities seemed to appear among schools, for example, in their capacity to organize remote learning activities. This is contrary to the basic concept of Finnish school policy, according to which all schools should offer identical quality.

Previous knowledge about the mechanisms of children's social exclusion allows us to assume that the closure of schools most negatively affected those children and young people whose risk of social exclusion was already very high and who did not receive adequate support at home. Current studies on school closure have identified possible consequences for children and families, such as economic problems for parents when they have to be away from work, loss of education, lack of school meals, social isolation, and possible harm to children's wellbeing and health, especially among the most vulnerable children [14]. There are reasons to direct supportive pedagogical measures toward them as well as analyze the possible adverse effects on children already at risk for exclusion.

### 1.2. Inclusive Policy, Inclusive Practices?

With regard to the concept of inclusion, it is worth mentioning the central role of special education in implementation of inclusion [15]. The Finnish strategy of special education "emphasizes inclusion over segregation and integration, and a pedagogical approach over a medical and psychological approach" to avoid diagnostic labeling [16]. As the principle of inclusive education embodies the Nordic welfare model, it can be seen as a common element of educational policy in all Nordic countries, although there is, in fact, a tendency to emphasize the economic utilitarian function of education [17]. Despite notable investments in prevention and alleviation of social exclusion caused by school dropouts at the secondary level, school dropouts have remained a focal challenge of schools in Finland [18], and the issue has become an important political subject [4]. In research, attention has also been paid to the mechanisms of implementation of the inclusion philosophy [19]. We do not know, so far, how these practices were shaped by the specific arrangements caused by the pandemic. It is obvious that the COVID-19 pandemic implicated several short- and long-term challenges. However, there was an urgent need and strong argument for keeping schools open as much as possible to promote children's wellbeing.

In the Finnish understanding of child welfare, well-educated professionals are urged to strive to identify possible problems and provide case-specific protection and support according to need. The particular situation with restrictions in social interaction understandably requires the communication and cooperation of professionals within the interdisciplinary teams and networks. Correspondingly, as cooperation between home and school is an important basic element of Finnish education [20] and highlighted as an inclusive practice [21,22], the move to remote school may have required the reorganization of communication practices between parents and school personnel. There is relevant research-based information available about parents' and professionals' perspectives on children's wellbeing and the quality of welfare work in elementary school [23] as well as institutions of early childhood education [24], but information is not yet available about the influence of the pandemic on cooperation between home and school. Furthermore, the role of technology in this cooperation is an understudied area [25].

### 1.3. Collaboration between Families and Schools

Co-operation between home and school is guided in Finland by law [26] and the national core curriculum [27], which gives schools the responsibility for building mutual networks with parents and supporting their parenting. In an inclusive educational system, parents are supposed to be active and respected members of the school society [11,28]. In general, collaboration with parents is considered an effective inclusive practice [22,29] and dialog an important part of enhancing decision making [30]. In the Finnish context, teachers regard parents as important peer collaborators and their most important contact. Parents, in turn, emphasize the importance of the child's social relationships in cooperation with teachers [31].

Barriers to communication usually appear between parents and teachers. The role of a teacher and his or her practices is crucial in the promotion of parental participation in education as respected members of school society [11]. In Finland, parents' participation seems to be more extensive in primary education with one class teacher than at the secondary level with several subject teachers

because of the differences in the structure of school participation and cooperation [32]. However, the position of parents seems to be constantly under negotiation [33]. There can be tensions between written instructions and reality, and teachers cannot always understand the family life context [34]. There are cultural differences that provide the possibility of misunderstandings, and the language used at schools can be difficult for parents [11]. On the other hand, especially in Finland, the usual medium for cooperation nowadays is digital communication through electronic tools, and face-to-face meetings have become rare [25]. It is common for parents to have smartphones with email and other applications, and they can communicate about their child's education when it best suits them [35]. Digital communication can create and maintain partnerships between a parent and a teacher and promote mutual feedback [25]. The meaningful use of ICT can strengthen the relationship between parents and teachers [36].

### 1.4. Resilience

At a time of challenging change, several elements of everyday life are not available in the mode we are used to. Families' weekly routines are connected to both work and school schedules, and in the case of school closure, both parents and children have to find new ways of organizing everyday life, work duties, and learning, as well as meals and free time. To maintain routines, an individual must determine how to adapt and find new solutions in the new situation. The concept of resilience is used to describe complex adaptation systems for conceptualizing the management of change in both individuals and institutions [37].

The concept of human resilience is defined in various ways [38]. It is defined as the mental and material resources of a person [39] and as the ability to "bounce back" after a challenging experience [40]. It is also seen as a dynamic process [41] and—more philosophically—as a way of being in the world [42]. In addition, resilience is seen as a process involving people's capacity to find resources to sustain their wellbeing, including the capacity of the environment to provide support [43]. It is understood as a personal quality [39], and it consists of the process of overcoming adversities and threats [44]. The role and quality of adversity vary in different definitions. Masten [45] sees resilience as implying the proper functioning of human adaptation and connects the phenomenon more to everyday living. Sarkar and Fletcher [38] connect resilience, in addition to daily challenges, to more crucial situations in one's lifespan, whereas Lutar, Cicchetti, and Becker [41] emphasize positive adaptation to challenges considered significant.

The concept of human resilience has mainly been analyzed in psychology [38], but it has also been researched in more social contexts, such as economics [46], as a capability of organizations [1] and environments [47]. However, most definitions of resilience combine two shared aspects: encountering adversity or risk and tending to adapt positively to the situation, and in general, resilience is seen as a process by which one adopts an affirmative position in a difficult situation [38].

For a psychological understanding of the concept of resilience, it is necessary in research to be sensitive and consider contextualization and sociocultural factors [38]. Both adversities and risks exist in individuals' relations to family, neighborhoods, institutions, and the larger sociohistorical context, which is at the same time the arena of wellbeing work focusing on children [48]. Resilience is described as a healthy social, community-based, and ecological system, and resilience in local communities requires the successful mobilization of resources and new orientations [49].

Resilience and the ability to enhance resilience on a regional level can be found in organizations, and the focus of research can be economical [46] as well as sociological [50]. Community resilience can be addressed according to four capacities: economic development, information and communication, social capital, and community competence [49]. In that sense, resilience is the capability of a system to overcome stress, and it is about transformation in a situation of change [50]. It can be understood as a method of governance to empower individuals, institutions, and other social and ecological systems to transform themselves [37]. Schools, as a part of communities, carry the meaning of continuity, and in times of crisis, there is the challenge of how to secure educational activities to support learning

and continuity in children's lives. In the context of schools, resilience is also understood as a goal of teaching [51]. All these dimensions correspond to descriptions and contexts used in the GEM [1].

On a practical level, some adaptive capacities to promote collective resilience are rather simple to implement at school. School-level practices like receiving and perceiving social support, enhancing a sense of community, and using flexibility and creativity in problem-solving [49] are easy to connect to inclusive practices [8] and a socially sustainable environment [1]. These capacities also have some points in common with coping strategies [52].

Furthermore, resilience research recognizes the concept of family resilience. We can extend the focus on the family as a functional unit, where relationships and key family processes may have an impact on the members of the family as well as on the family as a unit [53]. When thinking of sudden changes in the closing of the schools, a family—parents and children—needs to find a new way of coping with educational issues as well as economic and other challenges caused by lockdown in a society. Family resilience processes may be seen in how parents and children are able to "rally the system" under a crisis, buffer stress [53], and adapt to new routines that sustain security and continuity in everyday life.

A family resilience perspective can also be used to understand the interaction between school and families during the pandemic when the roles of school and family were challenged in supporting children's learning. According to Amatea, Smith-Adcock, and Villares [54], there are two basic premises to focus on with regard to family resilience in the context of school. The first premise emphasizes the key processes in a family that can mediate the impact of stressful crises and sustain their capacity to take care of and rear their children. The second premise is related to a family's capacity and processes for preparing their children to participate in school and how the school reciprocally responds to and strengthens these family processes. During the COVID-19 pandemic, the task of education was partly transferred to homes, and the crucial task of schools was to collaborate with children and parents and support their learning at home. Families had to take more responsibility, for example, for family learning opportunities, such as monitoring homework and children's school performance as well for family organizational patterns, which are key areas of family processes in supporting children's academic success [54].

## 2. The Context of the Research

The Finnish educational system had only a few days to reorganize lessons and teaching online. At the same time, parents had to find solutions for not only the organization of their children's education at home, but also their own work. In Finland, most women are active in work life. When remote school began, both families and teachers faced a complex situation. The national strategy in Finnish education was to jump to distance participation in services, and in education, this meant the use of e-learning. In remote learning, the parents' role and the nature of their support for their children changed greatly. The physical learning environment was now at home.

In Finland, a teacher has great autonomy when it comes to implementing the national curriculum. The curriculum provides shared content, and a teacher can decide how to teach it. The national curriculum that was deployed in 2016 also includes ICT as required content for all students [27]. Voogt and Pareja Roblin [55] describe different frameworks of competences needed in the 21st century, and in these frameworks, ICT has an essential role. According to them, ICT competence consists of information and technical literacies, but also ICT literacy, which means in a broad sense the skills needed for living in the modern knowledge society. For example, critical thinking skills are emphasized [56]. This has created a need for pedagogical ICT training for student teachers and in-service teachers [57]. When the transition to remote school occurred, training in the use of remote learning tools had not been extended to all teachers. Additionally, not all schools and classrooms were equipped with the necessary ICT. However, ICT had a crucial role in organizing remote school, because only the youngest school-aged children (first- to third-graders), as well as children with special needs, could study at

school. Remote schooling lasted for two months, and children returned to school to study for two weeks face-to-face before their summer holidays began.

According to Pöntinen and Räty-Zaborsky [58], parental involvement is associated with a child's learning when using ICT. The involvement is a part of so-called digital parenting, where a parent is simultaneously responsible for digital safety, digital literacy, and raising an ICT-competent future citizen [59]. Parents may or may not be aware of the benefits of technology [60]. Thus, if there is not a wide range of technology in the home, students might be unfamiliar with the use of ICT [61]. Even though ICT learning environments are rather similar to ICT used in parents' work, it must be noted that not all parents use ICT tools in their work, and unfortunately not all students and teachers were familiar with the web-based learning environments in which remote school took place. In the optimal case, parents had good e-learning competence, and they could participate in their children's remote school. Trust and Whalen [62] argued that during the COVID-19 pandemic, teachers did emergency remote teaching. Emergency remote teaching consists of ICT, but it also means preparing children for circumstances in which there is a lack of typical services like meals provided by schools, access to technology, and reliable internet.

The rapid change from schools to remote schooling at home challenged both schools and families to find new ways to continue teaching and support learning via digital devices. In this paper, we focus on the parents' views on how they accommodated to this rapid change and fostered a continuum of children's learning by distance in a sustainable way. Studying parents' views is important, because it will help schools support collaboration with families in the future in similar circumstances and promote the support of sustainable education for children. Keeping this in mind, we discuss sustainability as a framework for sustainable learning and education that connects individual and organizational capacity for intentional, proactive, and collaborative learning [63].

The research questions were therefore the following:

- How did families cope with remote educational practices during the COVID-19 pandemic?
- How did parents describe their resilient practices?
- How did families act if their resilient practices conflicted with those of remote education?

## 3. Methods

The data were collected via an online questionnaire at the beginning of the remote schooling period in spring 2020. An online questionnaire was distributed through social media using the snowball sampling method, in which participants are found by researchers as the first order zone. Then, later participants lead the network to second order zone participants and even further [64], but this time the network was online. The link to the electronic form written in the Finnish language was shared by the first author, a professional researcher in teacher education and special education, and the link was shared on Twitter 44 times. The link was shared by networks in other social media as well. According to the Finnish National Board on Research Integrity (TENK), it is not necessary to request ethical evaluation if adult, voluntary participants give written answers with their own consent in research and there is no expectation of harm to participants [65].

On the electronic form, parents were informed of the purpose of the research and that participation in the research was voluntary. Participating in research by using an electronic questionnaire was based on informed consent [66]. An online questionnaire enabled the respondents to remain anonymous, and a confidential and secure research process was guaranteed [67]. Participants in this study (see Table 1) had at least one child in compulsory education (aged 7–16 years).

**Table 1.** Participants in the study.

| Participants | | f (n) | f (%) |
|---|---|---|---|
| **Gender** | Female | 290 | 92 |
| | Male | 23 | 7 |
| | Do not want to say | 3 | 1 |
| **Employment status** | Employed | 230 | 73 |
| | Studying | 19 | 6 |
| | Suspended without pay during the COVID-19 | 8 | 3 |
| | Unemployed | 20 | 6 |
| | Retired | 4 | 1 |
| | Other * | 26 | 8 |
| | Several options chosen | 7 | 2 |
| | N/A | 2 | <1 |
| **Education** | Compulsory education | 3 | 1 |
| | Vocational or/and high school education | 73 | 23 |
| | Higher education | 239 | 76 |
| | N/A | 1 | <1 |
| **Number of children in Compulsory education** | 1 child | 148 | 47 |
| | 2 children | 129 | 41 |
| | 3 children | 29 | 9 |
| | 4 or more children | 9 | 3 |
| | N/A | 1 | <1 |
| **ICT-devices available at home \*\*** | computer | 284 | 90 |
| | tablet computer | 217 | 69 |
| | mobile phone | 306 | 97 |
| | only mobile phone | 8 | 3 |
| | no devices | 1 | <1 |

* such as entrepreneur, caregiver, nursing leave, sick leave, rehabilitation ** one family can have several devices, f > 316.

The e-questionnaire was open for one month, beginning the third day after school closure. We obtained 333 saved answers. There were five answers in which the family did not have children in compulsory education and nine answers in which the template was empty, one answer with no active consent, and two answers in which the participant stated that they were answering this questionnaire a second time—so the total analysis consisted of views from 316 parents.

The online questionnaire included both open and closed questions concerning the family background, ICT devices available at home (see Table 1), and parents' views on implementation of remote schooling. In open-ended questions, parents´ written responses varied from few words to multi-faceted explanations. In this sub-study, parents´ views were analyzed qualitatively from three open questions concerning worries, challenges, strengths, and solutions using an interpretative, deductive approach to categorize comparative content conceptions [64] and aiming to obtain the general sense of the information content [68]. A thematic criterion of the study consisted of adversities, capabilities, and practices that were formulated based on the body of research literature, and they provided a coding frame, where each unit equated to a theme [69]. A qualitative content analysis consisted of three phases: First, the second author interpreted and classified descriptions of adversities that parents experienced during their children´s remote schooling, and second, she analyzed parents´ descriptions of mental and material resources and capabilities both in and outside the family. Finally, the first author used the peer-debriefing technique [70] to scrutinize and refine the codes. The results of analysis were confirmed in the whole research group.

## 4. Results

### 4.1. Parents' Adversities during Remote Schooling

The rapid change in society, including remote school and remote work, was demanding for most of the respondents. Parents described diverse and complex adversities and worries caused by remote schooling. However, there was a clear group of 34 (11%) unequivocal answers of parents stating that they were not worried about their child during remote schooling. According to these parents, things were running smoothly within their own family. These parents also mostly praised the school for organizing a smooth transition to remote schooling.

> "Now there are no worries. Teachers of both children have understood the needs of e-learning: precise instructions and easy to contact during school days." (Employed, vocational and high school, two children)

Answers in this group were usually given without any specific descriptions. The answer was simply "Nothing" (to be worried about) (11 answers). The picture of worries and adversities consists of more descriptive texts. Thematically, there were concerns about children's learning and children's wellbeing, including social relations, managing daily life, and the use of ICT.

#### 4.1.1. Parents' Concerns about Children's Learning

One of the widely described adversities was parents' concerns about children's learning. Parents explained that they know their children and, therefore, they understand their children's diverse possibilities and risks in remote schooling well. Where some children worked relatively independently, other children needed more guidance and support to focus on studying and manage learning tasks. Parents shared their concerns about their children's abilities to focus on studying and stay motivated and to understand the tasks and how to work with them. Additionally, parents were worried about assessment, especially when the child was finishing compulsory education.

This unexceptional situation regarding remote schooling revealed differences in the quality of teaching. At the beginning of remote schooling, some parents reported that there was no teaching at all. Parents' experience was that the role of teachers just vanished, because children received only emailed lists of tasks. According to the parents, the quantity of school tasks did not fit the children's needs, tasks were poorly directed, or they arrived too late, and possibilities for contacting the teacher were too limited. Some parents felt helpless while trying to teach their children with unclear instructions. Simultaneously, some parents refused to take on a teacher's role at home. The wide spectrum of remote schooling practices was also confusing to parents and children. Some parents had specific opportunities to observe and compare the remote teaching of siblings. As one parent claimed, some teachers were fluent in remote teaching and producing video material, while other teachers just sent tasks (employed, higher education, two children).

> "Every school and every teacher have different [technological] systems in use. It's hard to keep up and keep children up." (Employed, higher education, one child)

> "School hasn't shifted to remote schooling in its exact definition. The assumption is that there is an adult at home who teaches the child. One teacher spoke correctly about home teaching while instructing the parents. —School hasn't taken responsibility in organizing the remote schooling even though all devices are available for it." (Employed, higher education, one child)

Many children needed extra support to manage remote learning, regardless of their age and capabilities. Parents experienced the need to help their children constantly, and even children who normally learned quite independently asked for help in this uncommon situation. Parents also felt that their own skills and knowledge were not sufficient to support their child's learning. Some parents even desired pedagogical skills to support their child's remote schooling.

"A 10-year-old child isn't ready to study alone without his parent regarding his abilities to learn remotely. He would need more support. It's also difficult for a child to understand that he should study as hard as at school, this isn't a holiday." (Employed, higher education, one child)

"Children don't have [now] support from their peers that they have at school." (Suspended without pay during COVID-19, vocational education, one child)

"My child, who normally acts excellent at school is continuously asking for help with more difficult tasks." (Student, primary education, two children)

Some parents took the responsibility of supporting their child's learning very seriously. These parents sacrificed their own wellbeing to help their child with homework during remote schooling. For example, one parent explained:

"The child couldn't do math tasks. I woke up at 6 a.m. during the weekend before the other family members woke up, to find advice from YouTube, Pinterest. I didn't receive any support from school. A child learnt; a mom got fatigued." (Employed, higher education, one child)

### 4.1.2. Parents' Concerns about Children's Wellbeing

The second problem seemed to be parents' concerns about children's wellbeing. School has a strong influence on the social life of children. Therefore, many parents were obviously worried about not only their children's learning but also their lack of social connections during remote schooling. Parents were worried about the loneliness of their child and the lack of social contact with their friends. On the other hand, staying at home caused more conflicts between family members. Parents described how the presence of a parent or a sibling disturbed the child's learning. Parents also claimed that children became nervous and angry with their parents more easily than with their teachers when facing challenges, because children may control their behavior more at school.

"A child is missing his friends." (Employed, vocational education, one child)

"A younger sibling disturbs [the child] a lot and we have tried to find activities for her." (Student, vocational education, seven children)

"Children protested on schooling definitely more to parents than to teachers (not overly courteous). We just needed to take a teacher position and abide by our statements." (Employed, vocational education, one child)

Parents also shared their concerns about excessive use of ICT devices. Since learning was organized remotely and the free time activities of some children were carried out via ICT devices, parents were worried about the physical health of their children, such as headaches and physical passivity due to a lack of outdoor activities. Along with physical health problems, some parents mentioned mental health issues, such as depression or anxiety, of their children in general or during the COVID-19 pandemic in particular. Parents recognized the inequality among families in their skills and opportunities to support their children's wellbeing.

"Ergonomics while working, watching the screen causes headache." (Student, vocational education, one child)

"[The challenge is] to take breaks and exercise. A child goes to school on foot or bike on a normal school day and she has recesses. [Now] a child would like to do all school stuff without breaks. We have solved this by having a small walk after breakfast. A child also has a regular "recess"." (Student, vocational education, one child)

> "Too much screen time for the fifth-grader since everything is done online." (Employed, higher education, two children)

> "Sometimes I have noticed anxiety and [a child] has asked about what would happen and "what if" questions. The situation raises a lot of anxiety not only for adults but also for others." (Employed, higher education, three children)

In Finland, school is, in many cases, the way to provide other services, such as health care. In this data, parents seldom described their worries about supporting a student. Few parents worried about how to meet and consult with personnel supporting learning and wellbeing, for example, speech- and physiotherapists and mental health service practitioners, during the COVID-19 pandemic.

### 4.1.3. Parents' Concerns about Managing Daily Life

Furthermore, parents had concerns about managing daily life. The pandemic not only affected schooling, but also changed the work environment of many parents to remote work. Children of different ages and parents began quickly to adapt everyday living and daily routines at home. Parents described how they managed various schedules and modified the home environment to fit the various needs of the family members. In practice, parents organized making lunch, caring for the younger siblings, relaxing, exercising, and even renovating their apartment simultaneously. Families with many children and single parents especially faced challenges in responding to the schedules and needs of every family member.

> "[The challenge is] the use of time. I do my own work in the mornings at 7–8:30 a.m., then I help my second-grader. After lunch it's time for the fourth-grader." (Student, vocational education, three children)

> "Doing my own work and guiding [the children] on schoolwork simultaneously is impossible and makes me nervous. I need to wake up early in the morning to work and continue late in the evening." (Employed, vocational education, one child)

Some parents claimed the lack of a peaceful place at home disturbed children while studying. For example, a parent described working in the kitchen while a teen did his homework for a home economics lesson. Remote school and work challenged families to find creative solutions in finding a space for everyone.

> "[We have] a lack of our own, peaceful space. Some of our children have studied [sitting] on a cold sauna bench." (Employed, higher education, two children)

Equipment at home did not always correspond with what the accomplishment of the children's homework would require. For example, one parent (unemployed, higher education, one child) described how the lack of white paper postponed homework until the parent visited the store.

### 4.1.4. Parents' Concerns about the Use of ICT

The use of ICT caused worries in parents. Even though Finland has a good reputation for enhancing ICT use in schooling, not all municipalities, schools, and teachers were ready for remote schooling. According to the parents, some teachers did not have work phones, so the only way for parents to communicate with teachers during remote schooling was via email or electronic platforms. Parents claimed that specific tools or applications were lacking not only from schools but also from home. Families with many children struggled especially with organizing ICT devices for all children and parents working remotely. Even though 90% of the participants had a computer and 69% a tablet computer (see Table 1), families did not have a personal device for every child and adult. Some schools lent devices to students. The problems were connected with incompatibility of ICT devices and specific applications that schools used for remote schooling or breaks and slowness of internet connection.

On the other hand, the lack of a suitable communication platform led some teachers and primary-school students to create groups in applications with age limits of 16 years, such as WhatsApp. Parents were also forced to create accounts for their young children so that they could participate in remote schooling.

> "We don't have enough ICT devices for remote schooling because (named electronic platform) didn't work on our tablet computer. Luckily, we could borrow an (named tablet computer) from school." (Employed, higher education, one child)

> "The web pages that a teacher recommends don't work with a slow internet connection." (Employed, higher education, three children)

Remote schooling required adequate ICT skills for children, parents, and teachers. Many parents claimed that it took time for all stakeholders to use ICT effectively. A child needed to learn quickly how to write a text, send it to a teacher, take photos, and communicate online. Additionally, many parents needed to learn ICT skills to support their children in remote schooling.

> "Using different applications has been challenging. I hardly use these applications at work so I can't [use them]." (Employed, vocational education, one child)

> "Writing on a computer is slow [for a child] because he hasn't gotten used to it. It takes so much time to do tasks: read the instructions, find the equipment, do the task, take photos of the ready-made task, and send a photo to a teacher." (Employed, higher education, one child)

> "[A child has] three different platforms in use to read instructions." (Employed, vocation education, two children)

During remote schooling, parents noticed that several teachers lacked basic ICT skills, which challenged teaching online. Some parents complained that schools did not provide any remote teaching, even though they had ICT devices for that.

> "Teaching was lacking 100%, a school does not organize any teaching, e.g., video teaching or other contact remotely even though everyone had a [digital] device that enables it." (Employed, higher education, one child)

> "The problem is the teacher, and the solution is that we contacted her by messages. We got a reply that the teacher is unable to use video contact with students. So there isn't any solution appearing to this problem." (Student, higher education, two children)

### 4.2. Capabilities and Resources during Remote Schooling

Even though the closure of the schools worried and strained many parents, almost all parents described elements that supported their everyday resiliency. The supportive positive elements that parents described referred to parents' personal capabilities as well as capabilities and resources inside and outside the family. There was also a small group of parents (n = 4) who claimed to have no strengths for surviving in supporting their child's learning during remote schooling. These parents explained that they were extremely tired, for example because of serious sickness, lost contact with the child, or simply did not know what to do to support the child in remote schooling. However, all of them had some notions of elements that supported them during remote schooling, such as taking outdoor exercise, having a child in special education, or mentioning that children were happy with remote schooling.

### 4.2.1. Parents' Personal Capabilities during Remote Schooling

On a personal level, many parents took a positive stance in facing the rapid change to remote schooling. They described being in a happy mood and getting excited about having something new in

their family life. They explained that they liked to teach their children and were satisfied by having an opportunity to do it more than before. Parents also valued education and learning and therefore were oriented to helping, understanding, and encouraging their children in remote schooling. Parents also named various skills that they utilized and described as their strengths during remote schooling, such as being persistent, calm, empathetic, and humorous. These characteristics supported the parents in trusting themselves and sustaining self-efficacy in a new situation. The change was not, however, easy for all parents. Some parents described their tiredness and anxiety over the COVID-19 pandemic and the changes that it caused to parents. Some parents as well as children also lacked motivation to study daily at home.

> "I am an active parent. I ask about things concerning school." (Employed, higher education, one child)

> "I have been interested in home schooling for a long time. I am good at inventing inspiring tasks and guiding a child with special needs in his teaching." (Employed, higher education, three children)

> "[My strengths are] calmness, playfulness, and empathy." (Employed, vocational education, one child)

> "It's boring!　An adult with a primary school child doing physical exercises." (Employed, vocational education, one child)

Parents' experience was that knowing about the school life helped them cope with remote schooling. Some respondents in this study had a direct connection to school because of their earlier or current work as a school assistant or a teacher. This direct experience was, nevertheless, not necessary for parents to help their children learn. Parents described having good general knowledge or ICT skills or having previous experience from home schooling. Parents retrieved more information and learned by themselves or together with the children.

> "I manage all study contents of the first-grader." (Unemployed, higher education, one child)

> "At the beginning we had technical challenges. We learned together to use applications, also siblings helped each other." (Student, education information missing, one child in primary school)

> "Remote schooling is already familiar to us because our oldest child got sick (allergic reaction) and studied at home." (Unemployed, vocational education, three children)

### 4.2.2. Families' Capabilities during Remote Schooling

At the family level, parents emphasized the importance of problem-solving and organizational skills in coping with remote schooling. Parents needed to react quickly to changing situations that required skills to anticipate and evaluate the changes. The urgent task was to schedule family life and manage the hustle of organizing remote work and remote schooling or—in the case of parents' work outside home—support the child's learning at home by distance. Additionally, if families had many children, parents were urged to arrange their schedule to support all the children. Parents emphasized the use of creativity and focusing on solutions and the use of common sense in solving problems caused by the new situations. Even though some families seemed to constantly suffer from rush and hustle, other families found comfort in structuring their daily life with routines and clear rules. In practice, parents focused on fixing mealtimes and paying attention to outdoor activities and exercise to balance physically passive remote schooling via ICT devices. Conversations among family members and clear, shared rules were appreciated in finding a balance in family life during remote schooling. Additionally, various memory aids were used to support the children and structure daily activities.

"I am organized and stick to routines. I'm able to handle daily life in exceptional times." (Parent on sick leave, higher education, two children)

"We have our own schedule even though the school recommends following the normal school timetable. We modified it according to parents' work, children's school entities, sunny day time, and children's resources. In practice, the first lesson starts at 9 a.m. during weekdays and the last one finishes at 7:30 p.m. We have breaks together to do outdoor exercises, play games, and mealtimes." (Student, vocational education, one child)

Many parents emphasized having a good relationship with their child that eased supporting the child's remote schooling. A good connection between parent and child is based on trust, love, and interest in the child's studying. Parents explained that they knew their child's strengths and personal characteristics. As one working, highly educated parent of four children noted:

"We understand children's individual needs and we want to support them individually." (Employed, higher education, four children)

Knowing the child allowed the parents to adapt daily routines according to the child's daily condition. In practice, parents allowed the child to follow his or her natural daily rhythm. Parents also paid attention to the daily mood and vitality of their child. One parent explained:

"I don't require too much from my child. I notice it fast if he gets tired and frustrated. A minimum is enough. I can apply my work [experiences] in services for people with disabilities that have benefited me in this situation. I value the balance of mind, not that we perform tasks [by force]." (employed, higher education, one child)

Parents shared their gratitude for skills and characteristics that their child had. Parents were pleased to see how children were very motivated in remote schooling and taking responsibility for their studying. For some children, studying seemed to be even easier than face-to-face teaching. Some parents also claimed that their child's age or lack of learning difficulties or basic diseases mitigated the child's learning at home.

"Children aren't loaded as much as normally and they aren't so tired all the time [which has] decreased crying and tantrums [and their] self-esteem has developed. They concentrate better on their tasks and enjoy learning." (Unemployed, vocational education, three children)

"Children are very talented in independent working. I give them an opportunity to study independently and I guide only if they or their teacher deliberately requires it." (Employed, higher education, two children)

Living arrangements of the family affected children's remote schooling. Some families lived in the middle of the forest or had a large yard that enabled outdoor activities and provided an environment for relaxing breaks. Some parents described how they had a large house where everyone found their own, peaceful corner to work or study. Additionally, some parents emphasized having materials and tools that supported their child's learning. In general, socio-economic status and particularly economic stability reassured parents.

"Both parents have a workplace and no fear of layoff, so it safeguards daily life significantly." (Employed, higher education, one child)

Parents expressed family and intimate relationships as a resource in supporting children's learning during remote schooling. A happy, strong family with a team spirit shared their concerns and distributed household work in a safe and constructive atmosphere.

"Everything is fine at home, it doesn't cause problems to stay in one apartment." (Unemployed, vocational education, one child)

"We have a good atmosphere in our family, and we can solve any problem." (Employed, higher education, four children)

### 4.2.3. Resources Outside the Family during Remote Schooling

Parents recognized resources outside the family during remote schooling. Parents' work played a significant role in many families. Parents asserted that remote work, that is, staying at home while remote schooling, eased their support of their child's learning. Parents were also relieved by a peaceful phase at work or an employer that understood the unusual situation.

> "I and my wife have the possibility to work remotely so we can help [the children] always when necessary and also control the children's participation in remote teaching." (Employed, higher education, two children)

> "A family-friendly workplace and the possibility to work remotely" [already before COVID-19]. (Employed, higher education, two children)

Some parents also referred to their networks in mastering family life during remote schooling. An ability to ask for help from these networks was highly appreciated among parents. Families received support from family members (such as older siblings and their spouses) as well as from their extended family and support networks (such as godparents and aunts). Grandparents were mostly mentioned as supporting their grandchildren's learning either online or at home. Even though the COVID-19 pandemic isolated many families from their networks because of restrictions in traveling, some families found a positive side of the situation. Some grandparents taught their grandchildren online or, as in the case in the next example of a caregiver, a highly educated parent of two children offered face-to-face support: A grandmother who lives abroad stays with us in quarantine because of the cancelation of her return flight and she can therefore help the children.

### 4.3. Parents' Views on Family Resiliency in Remote Schooling during the COVID-19 Outbreak

In general, parents had to find the balance between organizing their own duties and work and those of school for children in the home environment. The new situation of remote schooling meant a rapid process of adapting and finding resilience in everyday life and family functioning. Parents worried about their children's learning and well-being. The daily networks of children were changed during the COVID-19 outbreak, and parents worried about the lack of peer relationships and even the loneliness of their children. Remote schooling in the home environment also led to challenges in organizing the physical learning environment, particularly the provision of computers and desks for all family members, including both children and parents, if they worked at home (see Figure 1).

Parents' interpretations of the situation related to their interpretations of children's needs, and parents recognized and interpreted their own resources as well. The school was explicitly visible in relation to children's teaching, but parents described the shortage of the schools in contacting the parents. Therefore, parents were mostly dependent on their own resources.

We have focused on resilience from the perspective of parents' role as supporters and facilitators of children's learning at home during remote schooling. Despite this focus on family life and parenting, the results show multiple issues in organizing and coping with everyday life during the COVID-19 outbreak. Parents had to adjust their own activities and agency in a relatively short time within the family and in relationship to outer expectations from school and working life. Based on these results, the parents' ability to reflect on the situation and the needs of their children promote family resilience. The clear structure of days and routines created through parenting skills in the area of education promote coping. However, we do not yet know enough about other stressors in families and the long-term consequences of reconciling work, school, and family life during the COVID-19 outbreak.

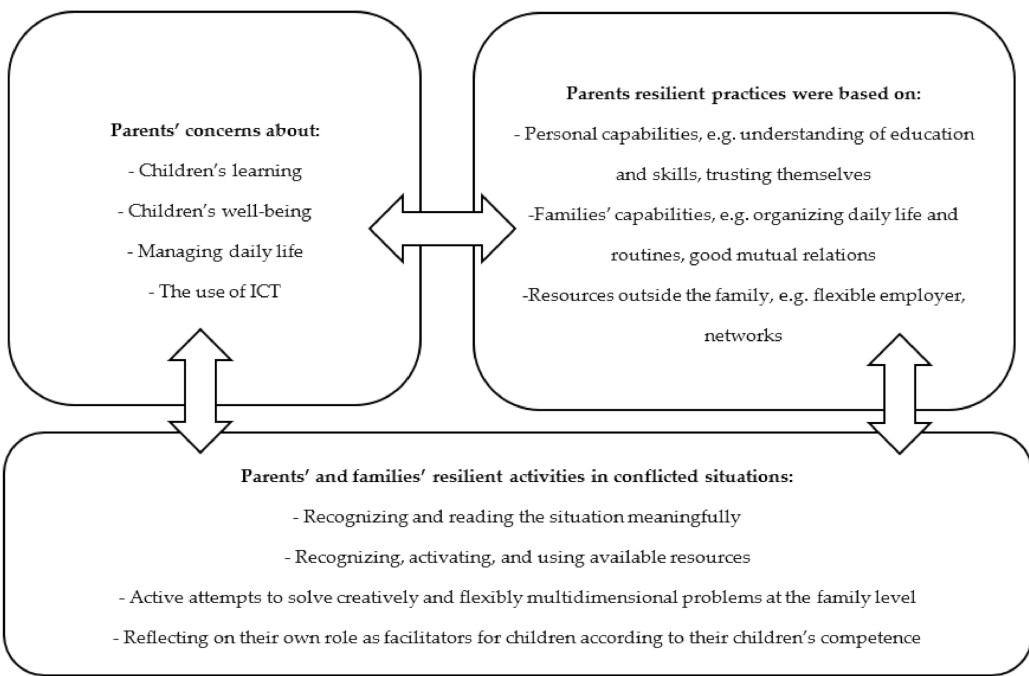

**Figure 1.** Parents' views on family resiliency in remote schooling during the COVID-19 outbreak.

## 5. Discussion

This study investigated parents' views on how they accommodated to the transition to remote schooling and sustained a continuum of their child's learning by distance in a sustainable way. We found this research very significant, because it reveals not only the resilience of parents and families, but also experiences of how the Finnish educational system functioned in the beginning of remote schooling because of the COVID-19 pandemic. This study supports earlier studies that have shown that resilience includes both personal and communal aspects [46], and all four aspects of collective resilience, i.e., community, communication, social capital, and economy [49]—or the lack of them—are experienced in parents' voices. There were practices supporting resilience, like fluent information sharing and reception and perception of social support in the school environment, but these are not to be taken for granted. Instead, resilient practices are dependent on schools or even on teachers. In a sustainable society, it is important to recognize individual- and family-based differences in needs and to be prepared to take care of all.

The results show that parents' experiences of their family resilience and the support that schools provided to children and parents varied greatly. Those parents who described having no worries concerning their child's remote schooling claimed capabilities in themselves and their families as well as in the schools that supported their child's remote schooling. This finding reflects the successful collaboration and dialog between parents and teachers to support effective and inclusive teaching practices [29,30]. More often, however, parents described having challenges and worries concerning their child's remote schooling. Parents also referred to challenges in their personal resilience, such as a lack of skills and knowledge to organize and support their child's learning at home. Additionally, a few parents referred to their networks when describing their resources and practices in coping with remote schooling. Since resilience is seen as a capacity to benefit from the environment [43], families could be supported in the future to collaborate more with each other and find support from their networks outside the school.

Remote teaching was experienced as qualitatively fragmented. Most parents' experiences were contradictory, including both good and challenging experiences. Parents' dissatisfaction with and criticism of the teaching is understandable, especially when they had a point of reference to how teaching could have been organized better.

From the parents' point of view, the observation of emergency remote teaching [62] revealed that situation was challenging and confusing for everybody. In the future, schools and teachers need to be prepared for shifting to face-to-face teaching, remote teaching, and flexible combinations of them to facilitate a sustainable continuum in children's learning and wellbeing. The contrast between different practices was clear, and it made it possible to require better practices for all students.

Furthermore, according to our research, ICT relates to everything. The fluent use of ICT was a clearly recognized resource both in families and at schools. The lack of cellphones for teachers is clearly a challenge for good cooperation. The school's role in supporting and strengthening family processes [54] and fostering good communication between families and teachers requires flexible solutions, such as smartphones [35]. Even if cooperation in general in Finnish society and schools is accomplished through electronic tools [25], there is a need to discuss more with parents about the decisions about which tools are chosen for communication. Promoting inclusive education requires smooth systems of communication, and it is a basic need to have a phone to use at work.

In families, the need for computers was demanding, especially if the same device was shared by remotely working parents and children. The ICT-based strategy pointed out the need of every municipality, school, and teacher not only to guarantee basic ICT remote-learning skills to teachers and children, but also to introduce ICT possibilities to parents and invite parents to share the use of ICT. A large number of parents managed their digital parenting [59] extremely well; however, there were also parents and children who struggled with basic ICT skills.

## 6. Conclusions

This research describes the fragmented field in education at the moment of a remarkable challenge. It addresses schools' and teachers' ability to promote resilience in families and to recognize differences in family and parental resilience. In times of change, a sustainable and inclusive society needs effective agency to support individuals [1]. Sustainable societies can create predictable strategies before the needs arise, and by doing so, enhance a sustainable and inclusive society for all.

Parents had to reflect on their role as facilitators of their children's learning in many ways. The role of a home teacher was rewarding when parents were able to organize support for their child based on the child's personal learning style and the daily rhythm in the family. On the other hand, parents experienced the limits of their own skills and resources in supporting the child. The tasks provided by the school varied from the lists of tasks to more elaborate learning via the internet. Parents had to adapt their own support for learning according to the teacher's instructions and the expectations of the school, as well as based on their own skills and consciousness of their child as a learner.

However, the context of remote schooling was confusing for many parents. They had an exceptionally high number of roles in the same environment, at home. They were parents and workers, they had to support their children, and, in some situations, they had to take on all the roles and duties of a teacher. Even if the worries were many and complex, there was the possibility of coping resiliently through knowing their child, helping the family to adapt, and developing positive routines, which were easier when school and teachers collaborated meaningfully. In addition, creativity, flexible living arrangements, and networks with social support seem to promote resiliency at the family level. A good understanding [12] and shared decision-making [11] between home, neighborhood, and school personnel is extremely important for enhancing participation and for promoting socially sustainable, resilient living environments [1]. The role of the school could be more effective and transparent, especially to promote collective resilience and a sustainable society. There are some limitations in interpretation in this research. The data were collected quickly after the closure of the schools to capture the early perceptions of parents during this exceptional situation. The short period of two months during spring 2020 was a kind of wide national experiment that tested family resilience in unprecedented times. This study reached parents who were able to invest time in answering the questionnaire. The conclusions based on the first results of the COVID-19 lockdown and

remote schooling experiences show that many families and parents are flexible and able to reflect the expectations for teaching in the home environment. We do not yet know the long-term consequences, and the information concerning harder-to-reach families is limited.

Social media was used to avoid exacerbating the problems of anyone in a stressful situation and to guarantee voluntariness. In this case, it meant that participants were mostly well educated. For example, academics [71] and the political elite [72] are active members of Twitter, so the decision to use social media affected the background of the participants reached. Obviously, the choice to use social media and its network algorithms, as well as the structure of the questionnaire, with open-ended questions, affected participant selection. A high percentage of participants in this research were female and well-educated people. However, in stressful situations, it is important to give participants the freedom to participate and answer or refuse involvement as much as possible, and social media was in this sense a meaningful solution, even if the written online questionnaire could not offer detailed content like an interview could. Further studies are thus needed to provide the parents an opportunity to share their thoughts and experiences by talking.

Additionally, the context of Finnish society must be noted when applying findings to other countries. The holistic way of organizing the education and welfare of children at schools in Finland, following the Nordic welfare system, differs from models applied in other countries. With regard to language, Finnish is gender-neutral, without distinguishing "he" and "she". Therefore, the sex of the children in the direct translations in the results section were generated by the authors in cases where the sex of the child did not appear explicitly in the parents' answer.

The SDG-inclusive educational system aims to serve and promote an inclusive society. Practically, the question is how to enhance participation for all, including parents, and to prevent all kinds of exclusion. The crucial question is how to enhance the flexible interplay between schools, welfare work served via education, and parents. By enhancing good strategies to promote inclusive participation for all families, education personnel, and welfare workers, it is possible to improve the sustainability of a society.

**Author Contributions:** Conceptualization: T.K., K.P., S.P.-H., R.V., and J.H.; methodology: T.K. and K.P.; validation: T.K., K.P., S.P.-H., R.V., and J.H.; investigation: T.K.; writing—original draft preparation: T.K., K.P., S.P.-H., R.V., and J.H.; writing—review and editing: T.K., K.P., S.P.-H., R.V., and J.H. All authors have read and agreed to the published version of the manuscript.

**Funding:** This research received no external funding.

**Conflicts of Interest:** The authors declare no conflict of interest.

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
