# Peer review of "Parents’ Views on Family Resiliency in Sustainable Remote Schooling during the COVID-19 Outbreak in Finland"

_sustainability, doi:10.3390/su12218844_

Round 1

Reviewer 1 Report

The authors idea of analyzing parents' perception of views on family resiliency in sustainable remote schooling during the COVID-19 outbreak in Finland) is welcome, necessary, and useful for the education system.

The authors' idea to analyze the perception of parents view on family resilience in sustainable remote schooling during the COVID-19 outbreak in Finland is welcome, necessary, and useful for the educational system.

Although the number of participants in completing the questionnaire is not very high compared to the number of parents who have children at school in Finland, it is still large enough to provide a useful picture.

I recommend the authors to review the results chapter. The way the results are processed must explicitly lead to conclusions. Even if the questionnaire also contained open-ended questions, they can be processed to justifiably show the opinion of the parents. A simple enumeration of some answers cannot lead to a conclusion.

After changing the way, the data is processed and presenting the results in a new form, the authors must present the modified conclusions.

Author Response

Dear Madam or Sir,

We thank for your review on our article manuscript. The feedback was constructive and supportive and helped us to improve our work. Accordingly, we made following revisions. We added subheadings to the results section to clarify the outcome of analysis. Furthermore, we provide a visualization of the qualitative analysis (See Figure 1 and chapter 4.3). As well we modified and developed conclusions (lines 761-769; 780-787)

We hope you find these revisions adequate. Once more, thank you for your thorough review.

Yours faithfully,

Teija Koskela et al.

Reviewer 2 Report

The topic of the article is currently a key issue.

Although the text is very extensive for what is said (especially in the introduction), I have to praise that the support of literature is great and everything is put into context.

For greater clarity of the article, I would recommend a more pronounced designation of respondents' citations from the rest of the text, and, in particular, I would add a summary visualization (conceptual graph).

Reading the article was interesting. I wish the authors great success with the article.

Author Response

Dear Madam or Sir,

We thank for your encouraging review on our manuscript. The feedback was supportive and helped us to improve our work. Accordingly, we did some revisions to the text. We added the figure 1 on page 15. It clarifies the results of the analysis. We also modified the quotations and used indentations and italic font.

We hope you find these revisions adequate. Once more, thank you for your thorough review.

Yours faithfully,

Teija Koskela et al.

Reviewer 3 Report

I miss the number of parents interviewed in the Mehods section.

In the Results section, create a summary table or graph containing the number of parents who were satisfied versus dissatisfied with distance learning.

Have you met families who did not attend distance learning at all?
Did you meet families who did not have an ICT device? You had information on whether the ICT device was lent to them or how the distance learning worked. Comment in the article.

Author Response

Dear Madam or Sir,

We thank for your encouraging review. The feedback was supportive and helped us to improve our work. Accordingly, we made following revisions.

The analysis was made based on online questionnaire and its written open-ended questions. This is now clarified, and word “written” is added on line 293 after Table 1.

In this data it is very difficult to classify parents’ satisfaction. Most of participant parents had very contradictory experiences: there were some practices or teachers´ actions that parents were satisfied with and simultaneously some other practices and teachers´ actions with what parents were not satisfied. This notion is added on the page 16, lines 734-735.

The information of devices in families is added in the Table 1.

Sentence “Some schools lent devices to students” was also added (p. 10, line 466).

We hope these revisions are adequate. Once more, thank you for your thorough review.

Yours faithfully,

Teija Koskela et al.

Reviewer 4 Report

Please notice that all comments are written in order to provide constructive feedback that takes into account the type of work usually published in Sustainability.

The paper cover a relevant and interesting topic, and provides a rich contextualisation that supplements the information obtained using a qualitative analysis of the responses obtained after using an online questionnaire. Although the examination procedure is sound and many examples and categories of analysis are provided, it should be noted that a very high percentage of the participants were female (92 per cent), and that a questionnaire of this type might not always be representative. In any case, and because the analysis could still be of interest due to the circumstances, I would recommend highlighting some of these methodological issues with the participants and the instrument a bit more strongly in the paragraph dedicated to describing some of the limitations of this study (page 14 of the manuscript).

Finally, I would also recommend including line breaks before and after each of the examples provided (in the words of the participants). By doing this, it might be possible to help the reader identify the different voices in the text while also improving the presentation of the manuscript.

Author Response

Dear Madam or Sir,

We thank for your encouraging review. The feedback was supportive and helped us to improve our work. Accordingly, we made following revisions.

We added justification concerning methodological issues, participants and chosen instrument on page 17, lines 781-800. As well we made quotations clearer by using Italic and indentations.

We hope that you find these revisions adequate. Once more, thank you for your thorough review.

Yours faithfully,

Teija Koskela et al.

Round 2

Reviewer 1 Report

I note favorably the intervention of the authors, noting the improvement of new version.